# Benchmarking the Effects on Human–Exoskeleton Interaction of Trajectory, Admittance and EMG-Triggered Exoskeleton Movement Control

**DOI:** 10.3390/s23020791

**Published:** 2023-01-10

**Authors:** Camila Rodrigues-Carvalho, Marvin Fernández-García, David Pinto-Fernández, Clara Sanz-Morere, Filipe Oliveira Barroso, Susana Borromeo, Cristina Rodríguez-Sánchez, Juan C. Moreno, Antonio J. del-Ama

**Affiliations:** 1Neural Rehabilitation Group, Cajal Institute, Spanish National Research Council (CSIC), 28002 Madrid, Spain; 2Systems Engineering and Automation Department, Carlos III University of Madrid, 28903 Madrid, Spain; 3Electronic Technology Department, Rey Juan Carlos University, 28933 Móstoles, Spain; 4CAR-UPM Associated Unit, Universidad Politécnica de Madrid, 28040 Madrid, Spain; 5Center for Clinical Neuroscience, Hospital Los Madroños, 28690 Madrid, Spain

**Keywords:** exoskeleton, human–robot interaction, electromyography, EMG control, exoskeleton control, benchmarking

## Abstract

Nowadays, robotic technology for gait training is becoming a common tool in rehabilitation hospitals. However, its effectiveness is still controversial. Traditional control strategies do not adequately integrate human intention and interaction and little is known regarding the impact of exoskeleton control strategies on muscle coordination, physical effort, and user acceptance. In this article, we benchmarked three types of exoskeleton control strategies in a sample of seven healthy volunteers: trajectory assistance (TC), compliant assistance (AC), and compliant assistance with EMG-Onset stepping control (OC), which allows the user to decide when to take a step during the walking cycle. This exploratory study was conducted within the EUROBENCH project facility. Experimental procedures and data analysis were conducted following EUROBENCH’s protocols. Specifically, exoskeleton kinematics, muscle activation, heart and breathing rates, skin conductance, as well as user-perceived effort were analyzed. Our results show that the OC controller showed robust performance in detecting stepping intention even using a corrupt EMG acquisition channel. The AC and OC controllers resulted in similar kinematic alterations compared to the TC controller. Muscle synergies remained similar to the synergies found in the literature, although some changes in muscle contribution were found, as well as an overall increase in agonist-antagonist co-contraction. The OC condition led to the decreased mean duration of activation of synergies. These differences were not reflected in the overall physiological impact of walking or subjective perception. We conclude that, although the AC and OC walking conditions allowed the users to modulate their walking pattern, the application of these two controllers did not translate into significant changes in the overall physiological cost of walking nor the perceived experience of use. Nonetheless, results suggest that both AC and OC controllers are potentially interesting approaches that can be explored as gait rehabilitation tools. Furthermore, the INTENTION project is, to our knowledge, the first study to benchmark the effects on human–exoskeleton interaction of three different exoskeleton controllers, including a new EMG-based controller designed by us and never tested in previous studies, which has made it possible to provide valuable third-party feedback on the use of the EUROBENCH facility and testbed, enriching the apprenticeship of the project consortium and contributing to the scientific community.

## 1. Introduction

We are witnessing an exciting era in which robotic technology for gait training is becoming a common tool in rehabilitation hospitals around the world. Advances in computing, materials, sensors, interfaces and manufacturing processes, along with the incorporation of basic knowledge of the neuro-physiological principles involved in motor recovery into intelligent controllers, are enabling better robotic rehabilitation services. Nevertheless, there is still no consensus on whether or not robot-assisted gait training benefits patients more than conventional therapy [1,2]. Moreover, the effectiveness of locomotor therapy is limited regardless of the training approach [3]. The use of ambulatory robotic exoskeletons, compared to robotic static gait trainers, may provide the patient with a more realistic and physiological gait condition, thereby increasing active participation in the therapy while providing task-consistent sensory and visual feedback.

However, the outcomes attained with ambulatory exoskeletons are still controversial. Published studies and reviews show considerable differences among protocols, targeted populations and variables analyzed, in addition to the specific differences among exoskeletons (number of joints, type of actuators and controllers, among others) [3,4]. Recent research claims that robot-assisted walking arises from the interaction between the human body, driven by the central neural system (CNS) through the muscles, neural loops, reflex mechanisms and the mechanical structure of each exoskeleton, driven by the controller through the joint actuators and sensors [5]. Although there is a growing body of knowledge that addresses the interplay between the neural and robotic structures in terms of user–robot interaction and further adaptation [6,7], there is still no consensus on the specific adaptation mechanisms, as well as what is the role of user preference on the performance of the human–exoskeleton system [8].

Further investigation of some of these physiological mechanisms, as well as involving user preferences within the control loop, would allow better identification of the patients who can benefit the most from robotic therapy, how to shape and customize the exoskeleton structure and control the functional needs of the patient, as well as designing a personalized exercise program to conduct with the exoskeleton, in order to maximize motor learning and, ultimately, recovery [9,10].

Along with the number and configuration of the joints, the main exoskeleton characteristic that affects human–robot interaction is how the robotic joints deliver torque to the human ones through the physical interface. Movement reference and joint actuator control are the two major areas of research. Several control algorithms for joint trajectory tracking have been proposed in the literature: from the simplest proportional-integrative-derivative (PID) control family, [11] to more sophisticated algorithms such as fuzzy control [12], robust variable structure control [13] and sliding mode variable structure control [14] have been used for lower limb exoskeleton robots [15]. A common feature of these algorithms is that they do not consider the wearer in the system besides limb inertia in some cases [16]. Exoskeleton–limb contact stiffness and user movement (either voluntary or reflex) greatly affect human–robot performance [17]. Therefore, a smooth and efficient movement might be achieved by focusing on the human–robot interaction instead of on the accuracy of the trajectory tracking, to achieve smooth and efficient movement [18]. Several control strategies have also been proposed that focus on the human–robot physical interaction: computed torque control [12] and different versions of impedance/admittance controllers [16]. However, despite the variety of control approaches investigated in the literature, the learning mechanisms of the user in response to robotic assistance are yet to be clearly established [19].

In order to improve human–robot interaction, muscle electromyography (EMG) has also been investigated as a predictor of human intention during walking [20,21], applied as a trigger for controlling prostheses and exoskeletons [22,23,24,25,26]. EMG signals show characteristic patterns of activation associated with each activated muscle in terms of onset timings, burst duration and levels of activation [27,28].

In addition, an analysis of combined muscle activation in terms of the number and characteristics of synergies has been proposed to provide a reliable representation of a person’s motor deficits and the degree of adaptability of their motor patterns [29]. It has been shown that the use of ambulatory exoskeletons does not alter muscle coordination, independently of the level of assistance [10,30,31].

EMG-based control algorithms may offer improved performance in terms of: (1) accuracy of movement selection, (2) intuitiveness and (3) response time of the control system [32]. Movement accuracy is relevant for achieving precise execution of a user’s intended task; an intuitive interface relieves the struggle of the user on the use of the control system; and, finally, the response time is important for avoiding any possible delay perceived by the user, which may hinder proprioceptive mechanisms. The patterns differ between healthy and pathological gait conditions and therefore can be used to assess improvements in muscle function, motor control and neuromuscular adaptations following rehabilitation interventions [33].

The main goal of this study was to compare the effects on human–exoskeleton interaction as well as user perception of the three main types of control strategies described above: joint trajectory tracking, joint mechanical admittance control and EMG-triggered control. We hypothesized that the EMG-Onset-triggered controller would improve human–exoskeleton interaction in terms of muscle coordination, physiological effort and walking experience compared to the admittance and trajectory controllers.

This exploratory study was conducted within the benchmarking initiative of the EUROBENCH European project, which developed the first unified benchmarking framework for robotic systems in Europe [34], comprised of a testing facility located at the *Center for Clinical Neuroscience* of *Hospital Los Madroños (Brunete, Madrid, Spain)* in Madrid (Spain), as well as a comprehensive set of testbeds with dedicated experimental protocols and performance indicators (PIs) [35] (hereinafter only testbeds). These results will allow companies and/or researchers to test the performance of their robots at any stage of development. In this work, we are therefore users of EUROBENCH’s project results. As shown in detail in Section 2, we designed our experimental protocol with the two testbeds most suitable for our objective, which are the EXPERIENCE [36,37] and PEPATO [38] testbeds: the EXPERIENCE testbed aims at evaluating the user’s physiological response and subjective experience during exoskeleton-assisted walking on a treadmill, while the PEPATO testbed aims at analyzing muscle coordination during exoskeleton-assisted gait. More details are provided in Section 2.

Therefore, the objectives of this work are twofold: (1) to compare the effects on human–exoskeleton interaction as well as user perception of the three main types of control strategies: joint trajectory tracking, joint mechanical admittance control and EMG-triggered control and (2) to provide third-party experience concerning the use of the EUROBENCH facility and testbeds, for both the project consortium and the scientific community.

## 2. Materials and Methods

As explained in Section 2.4, each protocol requires conducting exoskeleton-assisted walking tasks while wearing specific sensors to measure physiological data. Since the sensors required by each testbed do not interfere with each other, we combined both protocols in one single walking experiment. All instruments and methods applied in the experimental protocol are described in the following subsections.

### 2.1. Participants

Seven healthy volunteers (2 females and 5 males; 28 ± 7.14 years old; height of 172.28 ± 11.67 cm; weight of 66 ± 12.91 kg) participated in this study. Inclusion criteria included age between 18 and 70 years, ability to follow instructions, and understanding and signing the informed consent. Exclusion criteria included presence of any implanted electronic device; presence of ulcers or bedsores; occurrence of problems in lower limb joints in the past 3 months; history of previous surgeries in lower limbs in the past 6 months; presence of any pathology that affects movement; any other pathology such as cardiological, respiratory, renal, hepatic, oncologic or the like; taking oral anticoagulants; pregnancy; and not signing the informed consent form.

Participants were recruited through a call for participation sent by email to colleagues and by a notice posted in the facilities of Hospital Los Madroños (Madrid, Spain).

All participants were informed about the procedures and possible adverse effects and signed the informed consent form to participate. Both EXPERIENCE and PEPATO experimental protocols were approved by the Spanish National Research Council (CSIC) on 22 June 2021.

### 2.2. Instrumentation

#### 2.2.1. Exoskeleton

The exoskeleton used in this experiment was the Exo-H3 (Technaid S.L., Arganda del Rey, Spain). Exo-H3 has six actuators comprised of DC motors and harmonic gears at the hip, knee, and ankle joints (of both sides). Exo-H3 can connect to external devices via either CAN bus, Bluetooth, or WiFi, streaming exoskeleton’s parameters such as joint angles, interaction forces at foot, leg, and thigh exoskeleton sections, as well as foot–ground contact, at 100 Hz [39]. Each joint features both a trajectory and an impedance controller, which can be selected by the user. Joint angle references are fed to the controller from a walking kinematics database stored in memory. In this experiment, walking initiation/halt and velocity change was commanded for the exoskeleton through the CAN bus.

#### 2.2.2. Physiological Sensors

Following the requirements of the EXPERIENCE and PEPATO testbeds (see Section 2.5.2 and Section 2.5.3), electromyography (EMG), electrocardiographic (ECG), breathing rate (BR), as well as galvanic skin response (GSR) were recorded during walking trials. The devices used to record each of these measures were the following:**EMG signal**. A customized embedded processing unit that included an EMG amplifier and a voltage-controlled electrical stimulator (EAST, OT Bioelettronica, Turin, Italy) [40]. Surface electrodes Ambu^®^ WhiteSensor™ (Ambu^®^, Ballerup, Denmark).**ECG signal and BR**. A Zephyr BioHarness™ (Medtronic plc, Minneapolis, MN, USA) sensor was used. It is comprised of a fabric strap that incorporates the textile-type ECG electrodes and the breathing sensor. An electronic module placed at the strap acquires, converts and sends the ECG and BR data through Bluetooth.**GSR signal**. A Shimmer GSR+ Module (Shimmer, Dublin, Ireland) was used. It measures skin conductance between two electrodes attached to two fingers of one hand and converts and sends the GSR through Bluetooth data.

#### 2.2.3. Treadmill

Walking trials were performed over an instrumented treadmill (N-Mill, ForceLink B.V., Culemborg, The Netherlands) adapted from the C-Mill system from Motek (Motek, DIH Group, Houten, The Netherlands) with a belt speed that can be adjusted in steps of 0.01 m/s through the D-Flow software (Motek, DIH Group, Amsterdam, The Netherlands).

### 2.3. Control Strategies

#### 2.3.1. Trajectory Controller (TC)

The Exo-H3 features a PID position controller for each joint, which tracks joint trajectories from a walking kinematics database stored in memory at 100 Hz. After receiving the command encoding initiation, the system starts tracking the walking trajectory repeatedly, generating walking movement, until a command encoding stop is received. Figure 1 shows a conceptual diagram of the control scheme.

#### 2.3.2. Admittance Controller (AC)

The admittance controller available in the Exo-H3 was designed to increase actuation compliance, allowing for slight trajectory error. The magnitude of the error can be set externally through a constant value α, which corresponds to the percentage of movement error allowed by the controller for all the joints. Specifically, the controller compares the actual joint angle θ to the deviation defined by α, providing actuation proportional to the magnitude of the deviation. Figure 2 shows a conceptual diagram of the admittance control scheme. For the purposes of this work, α was set to 30% for all experiments. This value was obtained following a trial-error procedure in which we aimed at allowing tracking error yet having guidance towards the kinematic pattern. The value was in line with what was previously reported for the Exo-H3 exoskeleton in stroke survivors [41].

#### 2.3.3. EMG-Onset Controller (OC)

The EMG-Onset controller was independently designed to enable the user to trigger each step of the exoskeleton (Figure 3). This controller was adapted from the AC controller, where walking steps are continuously repeated to trigger the beginning of each step based on EMG activity from lower limb muscles (Soleus (*Sol*) and Rectus Femoris (*ReFe*), bilaterally). Right *Sol* and left *ReFe* are responsible for triggering the right step, whereas left *Sol* and right *ReFe* trigger the left step.

Specifically, the OC algorithm was developed upon two threshold-based methods: the single threshold (ST) and the double threshold (DT).They are based on the choice of a parameter level or threshold, which will serve as the boundary between the EMG signal amplitudes corresponding to muscle at rest (baseline) and contracting. The specific value (or values in the case of DT) of the threshold is calculated based on the amplitude characteristics of the EMG signal at baseline: the ST method [42,43,44] compares the raw signals to an amplitude threshold set from the mean power of the background noise. The main advantage of this method is that it allows one to directly use the raw EMG signal without processing, yet it is very sensitive to the choice of threshold.

On the other hand, the DT method [45,46,47,48] combines two different thresholds: amplitude (like the ST) and time, which gives robustness against false positives and improves detection accuracy. Nevertheless, due to the oscillating, high-frequency, nature of the EMG signal, the time threshold requires filtering the signal in order to work properly.

Given its reduced detection latency, the ST method was selected as onset detection, while the DT method was selected as offset (end of contraction) detection, where it was necessary to prioritize the robustness of the algorithm against false positives versus the decrease in latency at the time of offset detection. ST and DT methods were automatically applied on Sol and ReFe from both legs, to trigger each walking step commanded by the Exo-H3. The step is triggered when the onset method detects muscle onset in one muscle or the other. This redundancy can prevent the false negatives and increase the safety of the controller.

### 2.4. Experimental Protocol

EXPERIENCE and PEPATO protocols were used to compare the effects on human–exoskeleton interaction, as well as user perception, of the different control strategies described in Section 2.3. The EXPERIENCE protocol [36,37] aims at providing a benchmarking methodology for measuring both the user’s subjective perspective of the use of the exoskeleton by a newly developed multi-factor questionnaire and also to derive psycho-physiological indicators based on physiological data gathered during an exoskeleton-assisted walking test. EXPERIENCE testbed requires measuring ECG, GSR and the BR while walking.

In addition, the testbed requires each volunteer to sit for 4 min prior to walking trials (in this study, three different walking trials, each one to test the effects of each different controller) in order to have ECG, GSR and BR baseline data. Then, the exoskeleton-assisted walking begins, lasting 4 min while recording ECG, GSR and BR. After each walking trial, the user stops and sits back and is provided with the questionnaire, which was answered regarding the walking condition undergone. The EXPERIENCE protocol also includes custom-written software that processes and calculates performance indicators (PIs) based on the ECG, GSR and BR data, as well as on the user response to the questionnaires. More details on the specific PIs are provided in Section 2.5.

The PEPATO protocol [38] consists of custom-made software to evaluate the spinal locomotor output based on multi-muscle activity patterns, providing PIs of muscle coordination based on EMG signals recorded during exoskeleton-assisted walking. This requires measuring EMG of Soleus (*Sol*), Tibialis Anterior (*TiAn*), Rectus Femoris (*ReFe*), Vastus Lateralis (*VaLa*), Gastrocnemius Medialis (*GaMe*) and Biceps Femoris (long head, *BiFe*) muscles, as well as foot–ground contact of one leg. We used the EAST device described in Section 2.2 for measuring the EMG signals and extracted foot–ground contact from the exoskeleton data. PEPATO protocol does not provide the time or walking velocity constraints, so we used those suggested by the EXPERIENCE protocol. Details on the specific PIs of PEPATO are provided in Section 2.5.

#### 2.4.1. Experimental Setup

Figure 4 shows a schematic of the experimental setup that allowed us to combine both testbeds for measuring the data needed for each gait trial.

We used a PC/104 computer to synchronize exoskeleton data (joint angles, gait events) with EMG data from the EAST device. The exoskeleton was connected via CAN to PC/104, whereas the EAST was connected through a USB port to PC/104. We developed custom software written in Python that allowed us to manage data acquisition, storage and visualization, as well as to command the exoskeleton (walking initiation/halt and velocity, for TC and AC controllers, step trigger for the OC controller). In addition, the software also included the OC algorithm and a procedure for tuning the detection algorithm for each participant.

Figure 5 shows the front view (on the right) and the rear view (on the left), from two different subjects while they walked with Exo-H3.

#### 2.4.2. Study Design

In order to compare the effects of the three controllers (TC, AC and OC), we designed a cross-sectional, single-assessment, randomized experiment (Figure 6) in which data required for EXPERIENCE and PEPATO testbed were collected prior, during and after three walking trials. Each walking trial consisted of walking on a treadmill for 4 min with Exo-H3, using each of the three different controllers. The order of the walking trials was randomized for each participant.

For each participant, the experimental session started with the placement of GSR, BR, and ECG sensors on the chest and non-dominant hand. Then, the exoskeleton was placed on the user’s legs and adjusted for its specific leg length. EMG electrodes were then placed and the volunteer stood up and held a pair of crutches to check EMG quality. This was carried out by asking each participant to flex and extend each joint (ankle and knee) while visually inspecting EMG signals in real-time. After completing the instrumentation, the OC algorithm tuning procedure was conducted. To tune the algorithm, the user had to walk a minimum of two steps with the exoskeleton in passive mode—i.e., with the motor drivers disengaged—overcoming actuator resistance. During the procedure, the EMG signal of the *ReFe* and *Sol* muscles of both legs was acquired, visually evaluated and processed to define the most adequate onset thresholds. Specifically, the detection of the start of each step was estimated with the *ReFe* of the leg that starts the step or the contralateral *Sol* muscle.

After this, each participant sat back and rested with eyes closed for 4 minutes while GSR, BR and ECG data were recorded. These values served as a baseline to calculate physiological PIs. This was done before carrying out each of the three different walking trials in randomized order. Each walking trial (except OC) started with a familiarization period of 2 minutes. This allowed the user to get used to the changes imposed by each controller and also to set a comfortable walking speed. Regarding the OC walking trial, we had to define the specific threshold for each muscle (ReFe and Sol) before starting the familiarization period (see Section 2.3.3). After the 2 min familiarization period, there was a period of 4 minutes when each participant walked using each of the different controllers. Physiological (EMG, ECG, GSR, and BR), kinematic (joint angles from Exo-H3), and kinetic (foot–ground contact) data were recorded synchronously throughout the duration of each walking trial. After that, there was a 4 min resting period. Finally, at the end of each walking trial, each volunteer sat back in a chair and answered the EXPERIENCE questionnaire related to each condition tested. At the end of the session, participants also answered additional questions related to the general experience.

### 2.5. Data Analysis

#### 2.5.1. Exoskeleton

Exoskeleton joint angles (hip, knee and ankle from both sides) and foot–ground contact were sampled at 100 Hz. Joint angles for each walking step were normalized from 0 to 100% of the walking cycle, based on foot–ground contact forces. Maximum, minimum and range of motion (ROM) of each step were obtained and averaged across walking conditions (with each of the three controllers) and subjects.

#### 2.5.2. EXPERIENCE Testbed

As described in Section 2.4, EXPERIENCE aims at measuring both the user’s subjective perspective of the use of the exoskeleton through questionnaires and psycho-physiological indicators based on physiological data—ECG, GSR, and BR. The PIs can yield a maximum value of seven points, which are calculated from the answers to the questionnaire provided by the testbed (extracted from [36,37]):Usability: This is defined as the extent to which the exoskeleton can be used by the users to achieve specified goals with effectiveness, efficiency, and satisfaction in this specified context of use. High value of this PI indicates that the robot is highly usable.Acceptability: This relates to how the users perceive robots when interacting directly with them and how much you would be willing to introduce one into your everyday life. High value of this PI indicates that the robot is highly acceptable. This PI is comprised of four related constructs: attitude towards technology, self-efficacy, motivation, comfort, safety, and acceptability.Perceptibility: This evaluates the effects and influences that walking with the exoskeleton has on the user’s emotions, perceptions and quality of life. High value of this PI indicates that the robot positively influences emotion, perception and quality of life. The constructs associated with this PI are: embodiment and ownership, agency, emotion and attachment, health and quality of life.Functionality: This measures the perception of the characteristics of the exoskeleton in terms of ease of learning, the flexibility of interaction, reliability and workload. High value of this PI indicates positive features of the robot in terms of analyzed aspects. The constructs associated with this PI are: learnability, flexibility, robustness and reliability, workload, and functionality.

In addition, the PIs obtained from the physiological data were the following:**Stress**: This is defined as a state of mental or emotional strain caused by adverse circumstances. High value of this PI indicates that using the robot is stressful.**Energy expenditure**: This is defined as the amount of energy that is needed to carry out physical functions. High value of this PI indicates that using the robot requires high effort.**Attention**: This refers to the degree to which the user is consciously and continuously involved in the task. High value of this PI indicates that the robot use requires high attention.**Physical Fatigue**: This is defined as a type of distress generally conditioned by the exhaustion of one’s muscles due to the execution of a task. High value of this PI indicates that using the robot induces fatigue.

ECG was recorded at 250 Hz while GSR and BR were sampled at 25Hz. All PIs were calculated using the software available at the EUROBENCH facility at *Hospital Los Madroños*. Further details on the processing method and algorithms are available in [36,37]. The resulting PIs were averaged across walking conditions.

#### 2.5.3. PEPATO Testbed

As described in Section 2.4, PEPATO provides custom-made software for evaluating the spinal locomotor output based on multi-muscle activity patterns, providing PI of muscle coordination based on the EMG of eight muscles.

Nevertheless, the main activity of the knee flexor and extensor muscles is still captured by the measures of the VaLa, ReFE and BiFe. The PIs calculated with PEPATO software from the EMG are the following (extracted from [38]):EMG reconstruction quality.Full width at half maximum (FWMH): Estimated duration of basic patterns.Center of activity (CoA) of the basic patterns.

#### 2.5.4. Electromyography

EMG data were recorded at 2,000 Hz and low-pass filtered offline by a second-order Butterworth filter with a cut-off frequency of 6 Hz. The average EMG envelope of the four muscles used for the OC controller was obtained for each controller and normalized from 0 to 100% of the walking cycle. Average onset detection times of each step were obtained and averaged across subjects.

The number of onset detection per muscle was also obtained to highlight the importance of the muscles chosen in the actual control of the exoskeleton using the OC controller.

#### 2.5.5. Statistical Analysis

Non-parametric Friedman and post-hoc Wilcoxon tests with Bonferroni correction were used to test differences between walking conditions, and therefore between exoskeleton controllers. The non-parametric Friedman test is a non-parametric statistical test used to detect differences between different groups (in this case the three different controllers); it is preferable to use it when the same parameters are measured under different conditions on the same subject. The post-hoc Wilcoxon test is a non-parametric statistical hypothesis test used either to test the location of a population based on a sample of data or to compare the locations of two populations using two matched samples and the use of the Bonferroni correction can counteract the multiple comparisons problems. *p*-value was set to 0.05.

## 3. Results

### 3.1. Exoskeleton Kinematics

Figure 7 shows the average of the time-normalized joint angles for the ankle, knee and hip of both legs for all participants. It can be observed that knee and ankle angles for the TC condition were different than the angles obtained for AC and OC conditions.

Table 1 shows the average maximum, minimum and range of motion values of joint angles for the three walking conditions. We found statistical differences between TC and AC for all the variables and between TC and OC for all but the hip maximum. These results indicate that the users modified their walking pattern in both AC and OC conditions compared to TC conditions.

### 3.2. EMG-Onset Controller

As described in Section 2.3.3, the EMG-Onset algorithm is fed with the EMG from the *Sol* and *ReFe* muscles from both legs in order to estimate the step initiation and therefore trigger the exoskeleton step movement. Figure 8 shows the step-normalized and averaged EMG envelopes of these four muscles for the three walking conditions. Note that right *Sol* and left *ReFe* were used to trigger the right step, whereas the other muscles triggered the left, as previously explained.

Higher EMG activation was observed in both *Sol* for the OC walking condition compared to the TC and AC walking conditions. With respect to the *ReFe* muscles, the TC walking condition resulted in increased muscle activation compared to AC and OC conditions. These results suggest that the EMG-Onset algorithm required the user to increase the muscle activation of the *Sol* while decreasing the *ReFe* activation in order to trigger the steps.

Figure 9 shows the group-averaged EMG of the *Sol* and *ReFe* muscles for the OC walking condition along with the average onset detection for all steps and subjects. It is noticeable that no detection was obtained using the EMG from the left *ReFe* for any of the subjects and steps. Therefore, the proposed EMG-Onset algorithm shows robust performance due to a corrupt or even no EMG signal. In this case, it is shown how it was possible to successfully trigger the right step from the right *Sol* EMG signal.

The % gait cycle when onset onset detection obtained along a gait cycle is shown in Table 2 and is also illustrated by Figure 9. For the left step, right Sol activation was detected around 12 % of the gait and was never detected by the left ReFe. For the right step, both left *Sol* and right *ReFe* detected the beginning of muscle activation, around 60% of the gait.

The average values of onset detection for each muscle are shown in Table 3. Considering every onset detection for all subjects and trials, we can observe that the right Sol was the most used muscle, being responsible for almost half of the controller’s total onsets. Moreover, all the onset detections used to trigger the right step were achieved from the Right *Sol*. This could be due to the quality of the left *ReFe*, which presented noisy EMG and did not accomplish detecting EMG onset. Nevertheless, the PEPATO scenario PIs allow us to better understand the effects on muscle activation and coordination, as shown below.

### 3.3. Muscle Synergies (PIs Obtained from PEPATO Testbed Software)

With the six EMG muscles measured (previously defined in Section 2.4), we calculated the muscle synergies and their FWMH and CoA using the software provided by the PEPATO testbed. We configured the software to calculate four muscle synergies because it has been already reported that four synergies are enough to explain most of the EMG variability of the main lower limb muscles during gait [10,49,50]. Figure 10 shows the average synergy vectors obtained from the PEPATO testbed software for the three walking conditions in Table 4, which shows the FWMH (left) and CoA (right) for the four synergies.

The EMG reconstruction quality with four synergies (Table 4) was above 90% for all walking conditions. No statistical differences were found across walking conditions for all muscles within the four synergies.

Synergy 1 is mainly comprised of an ankle plantarflexion (*GaMe*) and knee extension activity (*VaLa* and *ReFe* muscles) with certain antagonist dorsiflexion activity (*TiAn*) for the TC walking condition. This activity is maintained for the AC walking condition and changes towards a knee antagonist co-contraction (*BiFe* vs *VaLa* and *ReFe* muscles) and increased ankle plantarflexion activity (increased contribution of the *Sol* muscle). The average mean duration of this synergy remains for the TC and AC walking conditions but shows a non-significant decrease for the OC (Table 4).Synergy 2 is mainly comprised of ankle dorsiflexion (*TiAn*) and knee extension (*VaLa* and *ReFe* muscles) for the TC walking condition. Similarly to Synergy 1, this activity mostly remains for the AC walking condition and changes in the OC walking condition towards ankle plantarflexion (increase in *GaMe* and *Sol*, reduction in *TiAn* contributions) and knee flexion (increased *BiFe*, reduction in *VaLa* and *ReFe* contributions). Similarly, the average mean duration synergy 2 remains for the TC and AC walking conditions, showing a significant decrease for the OC walking condition (Table 4).Synergy 3 shows a marked ankle plantarflexion (*GaMe* and *Sol* muscles) and knee flexion activity (*BiFe* muscle) for the TC walking condition. Similarly to Synergies 1 and 2, this activity remains with slight variations for the AC walking condition, but changes to a marked knee extension activity (decrease in the *BiFe* and increase in the *VaLa* and *ReFe* contributions), while ankle activity remains unchanged although a lesser contribution of the *Sol* muscle is observed. The average mean duration of this synergy remains for the TC and AC walking conditions but shows a non-significant decrease for the OC.Synergy 4 shows, for the TC walking condition, a noticeable ankle plantarflexion activity (*GaMe* and *Sol* muscles and a small *TiAn* contribution), whereas the knee shows an agonist–antagonist co-contraction (*BiFe* and *ReFe* muscles). Again, this activity remains with slight variations for the AC walking condition, whereas the OC walking condition shifts towards ankle dorsiflexion (increase in the *TiAn* contribution, decreasing in the *GaMe* and *Sol* muscles) with an increase in knee extension activity (*VaLa* muscle), although the contribution of the *BiFe* to co-contraction remains. Similarly, the average mean duration synergy 4 remains for the TC and AC walking conditions, showing a significant decrease for the OC walking condition (Table 4).

No significant variations in the CoA for the four synergies were found (Table 4).

As shown in Table 4, the OC walking condition resulted in less mean activation—FWHM—and delay—CoA—for all identified synergies, although not all walking conditions and variables were found to be statistically significant. Taken together, OC walking conditions resulted in a noticeable alteration of muscle coordination.

### 3.4. Subjective Perception (PIs Obtained from EXPERIENCE Testbed Software)

Table 5 shows the PIs obtained from the EXPERIENCE software averaged across subjects for the TC, AC and OC walking conditions. The PIs Acceptability, Functionality and Usability were rated above 4 out of 7, indicating that the users reported the exoskeleton and the associated walking condition as *positive*, *comfortable* and *safe*, *easy to use* and *robust*, and *highly usable*, respectively, according to the PI definitions provided in [36,37]. Regarding Perceptibility, the average value was below half of the scale, indicating that the users perceived the exoskeleton and the associated walking conditions as slightly negative emotion in terms of *embodiment*, *agency* and *attachment* [36,37]. No statistical differences were found across conditions; therefore, in terms of the user-perceived experience of use, users do not report differences across conditions.

Regarding the physiological-related PIs, the EXPERIENCE testbed provided an updated value of each PI for each walking minute. Figure 11 shows the four PIs for each condition, averaged across subjects, for four minutes of the walking trial.

No statistical differences were found for the Attention PI although OC walking condition required, on average, less attention than TC and AC conditions. With respect to Fatigue, also no statistical differences were found between walking conditions and across 1-min time intervals; all three walking conditions seemed to begin and end with a similar value of Fatigue but with little differences in the middle of the experiment. With respect to Energy expenditure, we found statistical differences between the first and fourth minute of the experiment (*p* < 0.05) for both the AC and TC walking conditions, in which a decrease in this PI can be observed. Despite the differences between walking conditions at the first and fourth minutes of the walking trial, no statistical differences were found for the whole trial. Lastly, Stress showed no statistical differences between walking conditions and across 1-min time intervals.

## 4. Discussion

This study focused on comparing the effects on human–exoskeleton interaction as well as user perception of three widely used exoskeleton control strategies—joint trajectory tracking (TC), joint mechanical admittance control (AC) and EMG-triggered control (OC)—while providing a third-party user-experience of the EUROBENCH testbeds, protocols and benchmarks. Our hypothesis was that the OC controller would improve human–exoskeleton interaction, in terms of muscle activation and coordination, and also reduce physiological effort while providing a better walking experience compared to the TC and AC assistance controllers. To test this hypothesis, we selected amongst the EUROBENCH available testbeds and protocols, the EXPERIENCE and PEPATO ones, which were combined and adapted to our objectives.

Overall, the admittance-based controllers—AC and OC—allowed the users to modify their walking kinematics, reducing the joint ROM (Figure 7 and Table 1), while also delaying the maximum and minimum of the curves. These results are consistent with other studies in which able volunteers walked at lower speeds using an exoskeleton when compared to self-paced slow walking speed (with no exoskeleton) [31,51,52]. The OC control strategy, while allowing the user to trigger each step individually, did not result in relevant kinematic alterations when compared to AC.

Regarding the synergy analysis obtained from the PEPATO software, we obtained an EMG reconstruction quality above 90% for all walking conditions, which indicates that four synergies were enough to account for the EMG variability for the three walking conditions. However, no statistical differences were found across walking conditions in any of the muscles within the four synergies. We hypothesize that a bigger sample size could have reached statistically significant differences in some of the synergy parameters, based on the trends observed.

We used a prototype EMG recorder (Section 2.2.2) designed initially for upper limb tremor assessment. This device uses pre-gelled single-use disposable EMG electrodes connected by cables to a connection board that is plugged into the device. This configuration might have a good performance for the target application—upper limb EMG monitoring in quasi-static configuration—but showed to be not adequate for EMG monitoring of walking with the exoskeleton. Firstly, walking is a dynamic task that produces constant movement and friction between cables, which might induce electronic noise coming from this constant cable movement and contact. Note that, although the electronic noise arising from that phenomenon can be relatively low, the cables translate a raw EMG signal which is also a very low-potential signal. In addition, the exoskeleton motors, drivers and power cables also produce considerable electronic noise within the cables. In order to minimize these effects, an EMG system that integrates measuring, filtering and analog-to-digital conversion close to the measuring point would be beneficial for obtaining high-quality and good signal-to-noise EMG signals.

The four synergies showed a similar composition for TC and AC walking conditions and showed a shift, not statistically significant, in the OC walking condition. With respect to TC and AC walking conditions, we found that synergies 1, 3 and 4 showed ankle plantarflexion activity and synergy 2 ankle dorsiflexion, whereas the knee extension was explained by the activity of synergies 1, 2 and 3. Synergy 4 showed knee flexion activity accompanied by some antagonist (extension) co-contraction from the *ReFe* muscle. These results align with what has been described in the literature. For example, Barroso et al. [50] also calculated four synergies, finding that ankle plantarflexion activity was described by synergies 1, 3 and 4, ankle dorsiflexion by synergy 3, knee extension by synergy 2 and flexion by synergy 4, with antagonist co-contraction of the *VaLa* muscle. Furthermore, Zhang et al. [53] investigated the effects on muscle synergies due to exoskeleton-assisted walking compared with free walking. Their results during free walking are slightly different, but also showed ankle plantarflexion activity in synergies 2 and 3, ankle dorsiflexion in synergy 1 and knee flexion in synergies 2 and 4 with considerable antagonist co-contraction in the latter (*ReFe* and *VaLa* muscles). Exoskeleton-assisted walking did not modify the synergies although some significant changes in the balance of the muscle groups were observed, i.e., changes in muscle activation while the overall effect of the synergy in the joint movement remained the same. However, synergy 4 showed an increment in the agonist–antagonist activation of the knee muscles. We also found this agonist–antagonist effect in the knee joint in synergy 4.

All four synergies decreased the FWMH duration (expressed as % of the gait cycle, Table 4) compared to the other two walking conditions. These results are consistent with a greater effort made by the user on the ankle plantarflexion and dorsiflexion movements in order to activate the onset algorithm. Note that estimation of the walking intention relied on the EMG from the *Sol* of the ipsilateral leg and the *ReFe* of the contralateral (i.e., loading) leg. Given the noisy quality of the EMG signals, the user had to increase muscle activation beyond what was natural to him/her to initiate the step in order to increase the EMG amplitude and to be successfully detected by the threshold algorithms. After the step detection, the user would decrease the overall muscle activation toward the level needed to complete the step. Note that the FWHM is a measure of the duration of the peak activation (i.e., the smaller the FWHM value, the higher the ability of the user to contract the muscle).

Regarding onset detection, we can see in Figure 9 that the right step was triggered, on average, in around 12% of the gait cycle, and the left step was triggered with both right *ReFe* and left *Sol* in around 60% and 62% of the gait cycle, respectively. The percentage of gait cycle timings obtained for both steps are in accordance with the values found in the literature [54].

Results also show that to start the left step, the algorithm detected, in all cases, muscle onset in the right *Sol*, not using the left *ReFe* as a trigger of the exoskeleton in any step. Despite the filtering stage, the signal from the *ReFe* muscle remained too noisy for the algorithm to work properly. Although the cables and electrodes were carefully revised, as well as the EMG readings, this problem could not be solved.

Differences observed in the kinematics and muscle activation were not reflected in differences in the overall physiological impact of walking (Figure 11). Furthermore, the group-averaged subjective perception did not differ across walking conditions (Table 5). Therefore, although the AC and OC walking conditions allowed the user to modulate their walking pattern and to adapt to the exoskeleton actions, these were not flexible enough to actually reflect a change either in the overall physiological cost of walking or in the perceived experience of use. Furthermore, the item that received the lowest score (*Perceptibility*, average 2.8 out of 7) indicates that the users experienced a low embodiment, agency and emotions (as defined in Section 2.5.2) during all the walking experiments.

Along with the already discussed limitations on the EMG recordings, another limitation of this study is its reduced sample size (*n* = 7 healthy subjects), which affects the likelihood of obtaining statistical differences and thus the generalizability of the results. However, the study allowed us to confirm the usefulness and limitations of our EMG-Onset control versus conventional AC and OC control.

## 5. Conclusions

This is, to our knowledge, the first study to benchmark the effects on human–exoskeleton interaction of three different exoskeleton controllers, including a new EMG-based controller designed by us and never tested in previous studies. Data collected will be very useful to improve our controller towards its application in incomplete spinal cord injury patients. Nevertheless, results indicate that, although no significant differences are observed, both AC and OC controllers showed potential to be explored as a gait rehabilitation tool. Furthermore, the INTENTION project was one of the first projects to perform experiments in EUROBENCH’s facility, which allowed us to give valuable feedback on the use of the testbeds and the overall facility, contributing to the improvement of the EURBENCH project and to the scientific community.

## Figures and Tables

**Figure 1 sensors-23-00791-f001:**
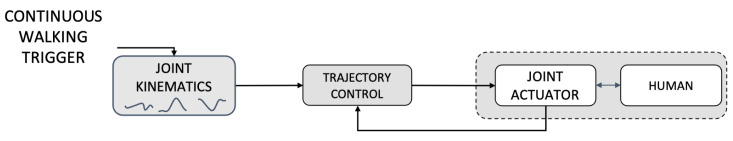
Conceptual diagram of the Exo-H3 trajectory control.

**Figure 2 sensors-23-00791-f002:**
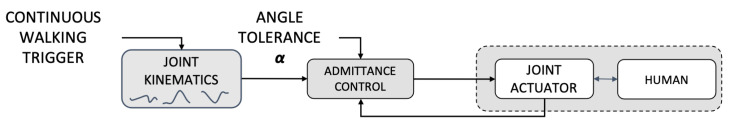
Conceptual diagram of the Exo-H3 admittance control.

**Figure 3 sensors-23-00791-f003:**
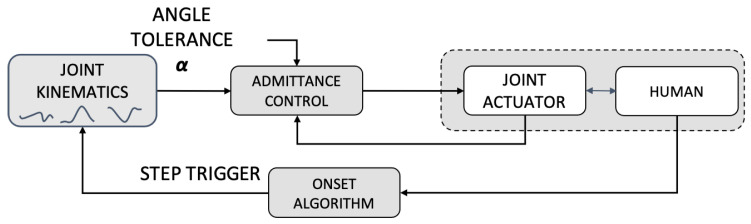
Conceptual diagram of EMG onset-based control loop designed for this experiment and implemented with Exo-H3.

**Figure 4 sensors-23-00791-f004:**
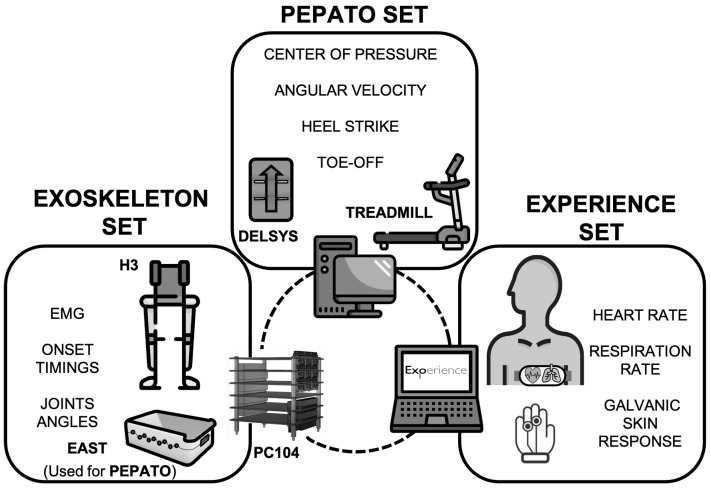
Experimental setup.

**Figure 5 sensors-23-00791-f005:**
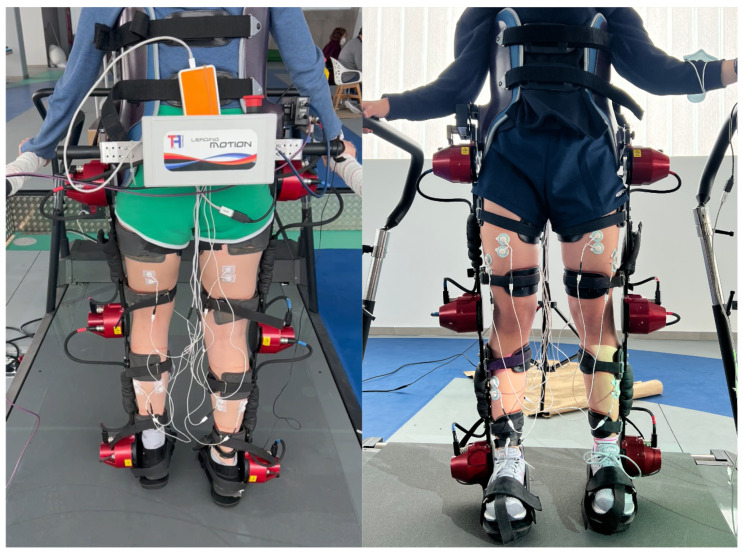
Pictures of two different subjects while they walked with Exo-H3. Most of the experimental setup is represented in the two pictures.

**Figure 6 sensors-23-00791-f006:**
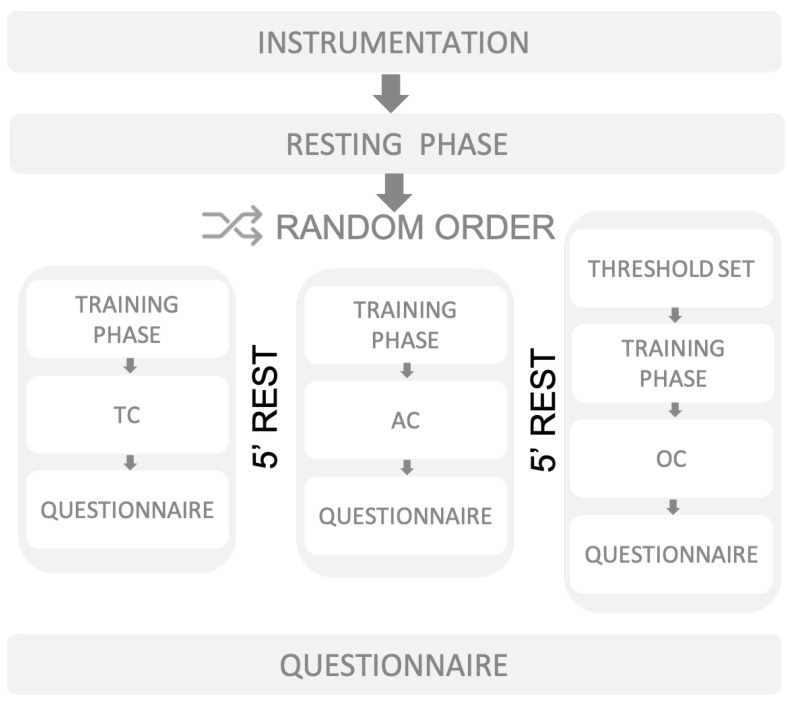
Schematics of the experimental protocol followed for each participant.

**Figure 7 sensors-23-00791-f007:**
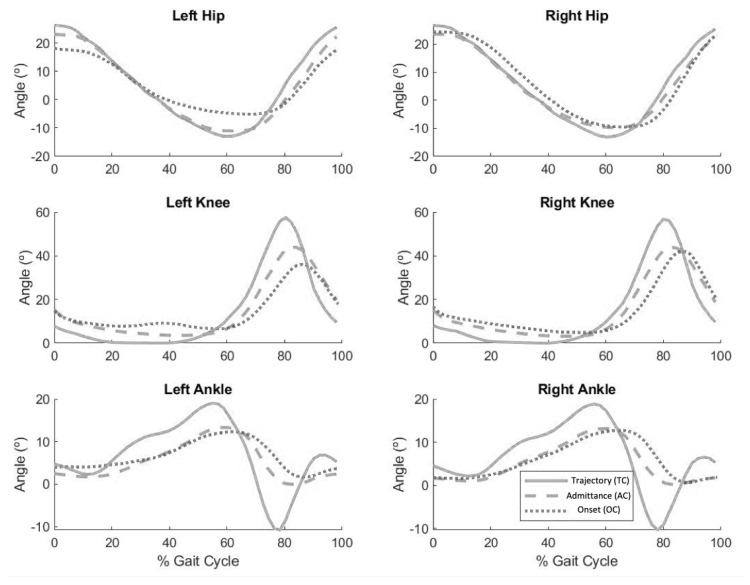
Average ankle, knee and hip joint angles when walking with TC (continuous), AC (dashed) and OC (dotted) controllers.

**Figure 8 sensors-23-00791-f008:**
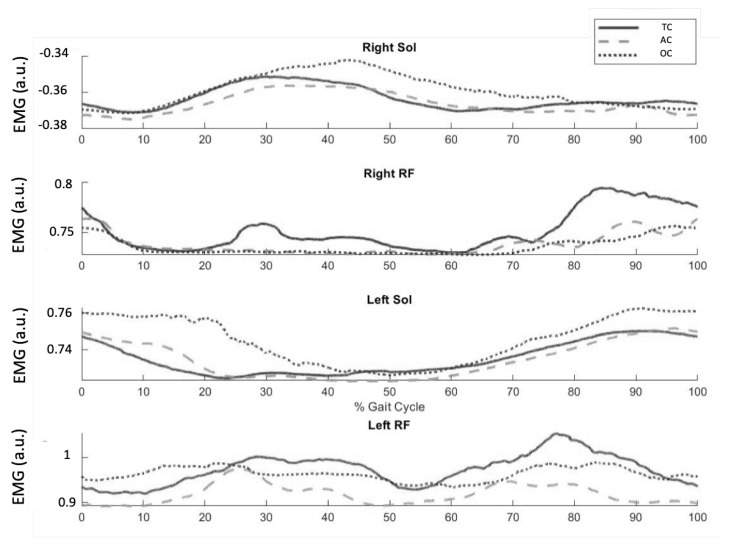
Normalized and cycle-averaged EMG envelopes of *Sol* and *ReFe* muscles.

**Figure 9 sensors-23-00791-f009:**
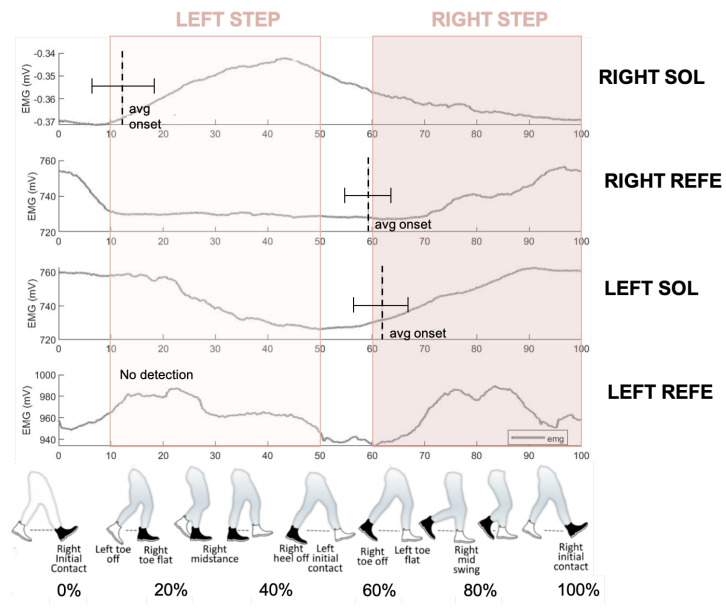
Average onset detection for the right and left *Sol* and *ReFe*.

**Figure 10 sensors-23-00791-f010:**
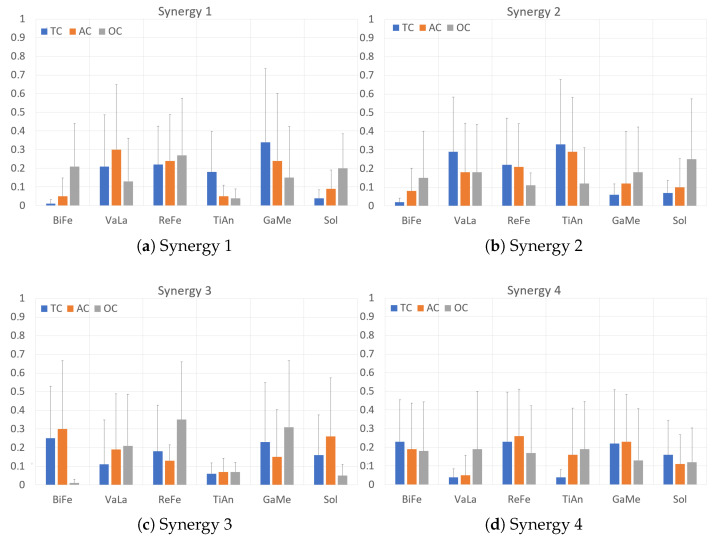
Average muscle synergies (right leg).

**Figure 11 sensors-23-00791-f011:**
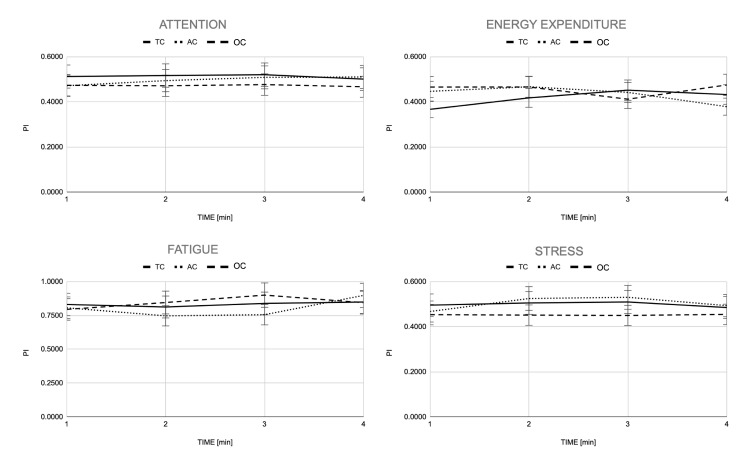
Average scores of Attention, Energy expenditure, Fatigue and Stress PIs for TC, AC and OC controllers. Scores of each PI were updated for each minute of trial.

**Table 1 sensors-23-00791-t001:** (Max) and (Min) angle and (ROM), of ankle, knee and hip joints for each tested controller. Positive values indicate knee flexion, hip flexion and ankle dorsiflexion. ^1^ statistical difference between TC and AC controllers; ^2^ statistical difference between TC and OC controllers; ^3^ statistical difference between AC and OC controllers.

		Controller				
		Mean	SD	Median	Max	Min
		TC				
Ankle	Max ^1,2^	19.40	0.32	19.43	19.56	19.31
	Min ^1,2,3^	−12.97	0.12	−12.97	−12.92	−13.01
	ROM ^1,2^	32.40	0.40	32.40	32.57	32.23
Knee	Max ^1,2^	60.45	0.89	60.55	60.64	60.46
	Min ^1,2,3^	−0.03	0.14	−0.03	0.01	−0.07
	ROM ^1,2^	60.48	0.87	60.58	60.63	60.53
Hip	Max ^1,2^	27.20	1.81	27.11	27.12	27.10
	Min ^1,2,3^	−13.53	0.80	−13.54	−13.48	−13.60
	ROM ^1,2^	60.48	0.87	60.58	60.63	60.53
		AC				
Ankle	Max ^1,2^	14.65	1.68	14.86	14.93	14.79
	Min ^1,2,3^	−2.92	3.32	−3.41	−3.22	−3.60
	ROM ^1,2^	17.58	4.97	18.28	18.54	18.01
Knee	Max ^1,2^	49.02	4.11	49.56	49.56	49.56
	Min ^1,2,3^	2.67	1.90	2.44	2.69	2.18
	ROM ^1,2^	46.35	5.48	47.13	47.38	46.87
Hip	Max ^1,2^	24.68	2.60	24.52	24.73	24.31
	Min ^1,2,3^	−11.26	2.07	−11.77	−11.13	−12.40
	ROM ^1,2^	35.94	3.32	36.30	37.15	35.45
		OC				
Ankle	Max ^1,2^	14.04	1.49	14.17	15.62	13.72
	Min ^1,2,3^	−1.45	2.26	3.69	3.85	3.52
	ROM ^1,2^	44.17	4.04	44.63	44.69	44.56
Knee	Max ^1,2^	47.91	2.67	48.31	48.54	48.08
	Min ^1,2,3^	3.74	2.62	3.69	3.85	3.52
	ROM ^1,2^	44.17	4.04	44.63	44.69	44.56
Hip	Max ^1,2^	25.06	2.61	24.47	24.76	24.19
	Min ^1,2,3^	−11.00	1.77	−11.28	−10.91	−11.65
	ROM ^1,2^	36.06	2.95	35.76	35.85	35.68

**Table 2 sensors-23-00791-t002:** The % gait cycle when onset of muscle activation was detected for each muscle.

Right Step		Left Step	
Right ReFe	Left Sol	Left ReFe	Right Sol
59.24 ± 4.35%	61.74 ± 5.22%	No Detection	12.18 ± 5.92%

**Table 3 sensors-23-00791-t003:** Average onset detection percentage in the four control muscles.

Right Step		Left Step	
Right ReFe	Left Sol	Left ReFe	Right Sol
17.4%	32.85%	0.00%	49.74%

**Table 4 sensors-23-00791-t004:** PIs provided by PEPATO testbed. See Section 2.5.3 for the description of each PI. Data are averaged across walking conditions. FWHM and CoA are expressed as % gait cycle. ^3^ statistical difference between AC and OC controllers.

		PEPATO PI								
		EMG Reconst. Quality	FWHM 1	FWHM 2 ^3^	FWHM 3	FWHM 4 ^3^	CoA 1	CoA 2	CoA 3	CoA 4
TC	Mean	0.95	16.07	18.57	22.21	14.79	16.74	31.49	57.61	57.25
	SD	0.03	8.12	15.54	13.95	12.98	11.06	9.09	29.55	39.98
	Median	0.94	14.00	14.50	23.00	13.00	15.34	30.23	69.18	76.50
	Max	0.987	32.50	50.50	44.00	35.00	30.02	48.76	97.39	94.49
	Min	0.90	6.50	6.00	6.50	0.00	3.39	16.37	6.79	0.07
AC	Mean	0.94	15.07	18.21	19.50	26.86	31.40	46.44	59.18	51.27
	SD	0.05	11.87	6.81	16.27	12.80	36.21	24.04	17.45	37.91
	Median	0.95	16.50	19.50	26.50	24.50	18.51	37.96	64.70	59.82
	Max	0.98	29.00	29.00	40.50	43.00	99.78	94.59	80.07	96.19
	Min	0.84	0.00	6.50	0.00	6.50	3.79	22.17	31.61	7.60
OC	Mean	0.95	6.29	6.29	7.14	5.29	25.66	36.38	55.49	67.26
	SD	0.03	12.89	4.94	7.81	6.64	31.35	21.41	18.45	29.50
	Median	0.95	0.00	5.00	3.00	1.00	20.26	46.55	54.06	73.56
	Max	0.99	35.00	16.00	18.00	17.50	88.89	59.14	87.24	92.34
	Min	0.90	0.00	1.00	0.00	0.50	0.01	5.51	33.95	7.74

**Table 5 sensors-23-00791-t005:** Average and SD of questionnaire-related PIs for all subjects and walking conditions. The PI can provide values between 0 and 7 [36,37].

		EXPERIENCE PI			
		Acceptability	Funcionality	Perceptibility	Usability
TC	Mean	4.63	3.97	2.85	4.38
	SD	0.35	0.65	0.30	0.35
	Median	4.60	4.06	3.37	4.23
	Max	5.40	4.67	4.69	4.75
	Min	3.87	3.09	0.00	4.17
AC	Mean	4.63	3.96	2.84	4.40
	SD	0.36	0.52	0.33	0.46
	Median	4.63	4.06	3.36	4.17
	Max	4.63	4.83	4.63	5.00
	Min	4.63	2.89	0.00	4.03
OC	Mean	4.63	3.95	2.81	4.43
	SD	0.36	0.56	0.40	0.47
	Median	4.60	4.03	3.39	4.20
	Max	5.40	4.83	4.49	4.96
	Min	3.87	2.91	0.00	4.13

## Data Availability

The datasets used and/or analyzed in this study are available from the corresponding author upon reasonable request.

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
