# Peer review of "Benchmarking the Effects on Human–Exoskeleton Interaction of Trajectory, Admittance and EMG-Triggered Exoskeleton Movement Control"

_sensors, 2023, doi:10.3390/s23020791_

Round 1
Reviewer 1 Report
Below is my feedback
Introduction
The introduction is well written.
Methodology
The methodology is well written, but needs additiona information to increase replicability.
1. How were participants recruited?
2. What was the inclusion/exclusion criteria?
3. How did you come up with 7 participants? Did you perform an a priori power analysis?
4. What were the demographic characteristics of these participants?
5. On line 316- Please label this as "Physical Fatigue"
Results
On line 385 you stated that you used 6 EMG muscles. I went back and re-read your manuscript multiple times and could only find where you descrbed using 4. Please clarify
Although I can't tell whether your data is normally distributed or not, if you are going to use non-parametric analyses (which is the correct analysis for this particular study), I would highly recommend using median and range in your tables.
Otherwise the results are well written and the tables and figures are very useful.
Discussion
What are the limitations of this study?
Please provide a conclusion paragraph.
Author Response
We want to thank the reviewers for their comprehensive feedback and recommendations which were very helpful in improving our manuscript. In an effort to address those comments, we have provided more information to complement the previous version of the manuscript. Specifically, we expanded section 2 (‘Materials and Methods) with information that was missing and might help increase the reproducibility of methods, as well as additional information in the ‘Abstract’, ‘Introductions’ and ‘Conclusions’ (new section added to the manuscript) sections, aiming to highlight the significance and main contribution of this work. Changes in the new version of the manuscript with respect to the previous one are highlighted in yellow. Please see below our responses.
Reviewer #1
R.1: The methodology is well written, but needs additional information to increase replicability.
R1.1: How were participants recruited?
Details on the recruitment process have been added in lines 138-139: “Participants were recruited through a call for participation sent by email to colleagues and by another call posted in the facilities of Hospital Los Madroños (Madrid, Spain).”, where the experiments were carried out.
R1.2: What was the inclusion/exclusion criteria?
Inclusion/exclusion criteria were added to the manuscript (lines 129-137): “Inclusion criteria included age between 18 and 70 years; ability to follow instructions; understanding and signing the informed consent. Exclusion criteria included presence of any implanted electronic device; presence of ulcers or bedsores; occurrence of problems in lower limb joints in the past 3 months; history of previous surgeries in lower limbs in the past 6 months; presence of any pathology that affects movement; any other pathology such as cardiological, respiratory, renal, hepatic, oncologic or similar; taking oral anticoagulants; pregnancy; do not sign the informed consent.”.
R1.3: How did you come up with 7 participants? Did you perform an a priori power analysis?
Although we formulated hypotheses and power analysis is directly related to tests of hypotheses, our study was meant to compare different types of exoskeleton control strategies. Given that our EMG-Onset stepping controller was never tested under such circumstances and these testbeds had not been applied to benchmark the other two controllers, we had no prior references to help calculate the required sample size. That being said, we acknowledged the sample size as being a limitation of this study (see lines 573-577 in the ‘Discussion’ section) and clarified that this is an exploratory study in line 8 (‘Abstract’) and line 103 (‘Introduction’).
R1.4: What were the demographic characteristics of these participants?
Information on the demographic characteristics of the participants was given from line 129 to line 131: “Seven healthy volunteers (2 females and 5 males; 28 ± 7.14 years old; height of 172.28 ± 11.67 cm; weight of 66 ± 12.91 kg) participated in this study”.
R1.5: On line 316- Please label this as "Physical Fatigue".
As recommended by the reviewer, the PI labeled as ‘Fatigue’ in the previous version of the manuscript is now labeled as ‘Physical Fatigue’ throughout the manuscript.
R.2: On line 385 you stated that you used 6 EMG muscles. I went back and re-read your manuscript multiple times and could only find where you described using 4. Please clarify.
In section 2.4. (‘Experimental Protocol’), we listed the 6 muscles where EMG was recorded in order to calculate muscle synergies using PEPATO testbed: “It requires measuring EMG of Soleus (Sol), Gastrocnemius Medialis (GaMe), Tibialis Anterior (TiAn), Rectus Femoris (ReFe), Vastus Lateralis (VaLa) and Biceps Femoris (long head, BiFe) muscles, as well as foot-ground contact of one leg.”. This is the same list of muscles that we referred to in section 3.3. (Muscle synergies) where we present the results of this PI (line 410). On the other hand, the list of 4 muscles that the reviewer mentioned are the muscles used to feed the EMG-Onset algorithm (Sol and ReFe from both sides, see lines 194-199 and 380-385 in sections 2.3.3 and 3.2, respectively).
R.3: Although I can't tell whether your data is normally distributed or not, if you are going to use non-parametric analyses (which is the correct analysis for this particular study), I would highly recommend using median and range in your tables. Otherwise the results are well written and the tables and figures are very useful.
Thank you for this relevant suggestion. We have now expanded the tables of this study (Table 1, Table 4, and Table 5) to also present the median and range (including minimums and maximums) for all variables.
R.4: For the Discussion, what are the limitations of this study? Please provide a conclusion paragraph.
We have included a new paragraph in the ‘Discussion’ section (see lines 573-577) clarifying the main limitations of the study.
Reviewer 2 Report
This study focused on comparing the effects on human-exoskeleton interaction as well as user perception of three widely used exoskeleton control strategies. The topic of this paper is closely related to the hot topics in the field of exoskeleton, and content is substantial. But there are a few things that need to be improved.
1. In the abstract and conclusion, the paper does not highlight the new discovery, significance and contribution, and the difference and innovation from other similar work;
2. In 2.1, if relevant regulations are not violated, the gender, age, physical quality and other information of the seven healthy volunteers should be shown, because these factors may affect the test results;
3. In 2.2.2, what is the effect of EMG signal, ECG signal, BR and GSR signal data collected by various physiological sensors on the main work of this paper, and what is the relationship between them and the PIs in the following paper. These questions should be further explained;
4. In 2.3.2, what is the basis for the allowed error value α to be set at 30%? Does this value have any influence on the subsequent comparison? Further explanation of these issues should be given in the paper;
5. In 2.5.5, "Non-parametric Friedmann and Post-hoc Wilcoxon tests with Bonferroni correction" and "P-value" are not mentioned in the previous article. The concepts and professional terms mentioned first time in the paper should be explained. What is the function and setting basis of P-value;
6. In this paper, the exoskeleton joint data and human physiological data under the three control strategies are compared. However, if the above data under the condition of natural gait (with no exoskeleton) are added to the comparison, the analysis of the results of this paper will be more meaningful.
Author Response
We want to thank the reviewers for their comprehensive feedback and recommendations which were very helpful in improving our manuscript. In an effort to address those comments, we have provided more information to complement the previous version of the manuscript. Specifically, we expanded section 2 (‘Materials and Methods’) with information that was missing and might help increase the reproducibility of methods, as well as additional information in the ‘Abstract’, ‘Introductions’ and ‘Conclusions’ (new section added to the manuscript) sections, aiming to highlight the significance and main contribution of this work. Changes in the new version of the manuscript with respect to the previous one are highlighted in yellow. Please see below our responses.
Reviewer #2
This study focused on comparing the effects on human-exoskeleton interaction as well as user perception of three widely used exoskeleton control strategies. The topic of this paper is closely related to the hot topics in the field of exoskeleton, and content is substantial. But there are a few things that need to be improved.
We thank the reviewer for the positive appraisal of our exploratory study, as well as for the valuable comments and suggestions. Please find our detailed responses to the comments / suggestions below.
R.1: In the abstract and conclusion, the paper does not highlight the new discovery, significance and contribution, and the difference and innovation from other similar work.
We have included a conclusion section (from lines 578 to 588), highlighting the significance and contributions of our work. We have also updated the abstract accordingly (From lines 20 to 27).
R.2: In 2.1, if relevant regulations are not violated, the gender, age, physical quality and other information of the seven healthy volunteers should be shown, because these factors may affect the test results.
This is a pertinent observation. The demographic information on the volunteers is now described between lines 129 and 131.
R.3: In 2.2.2, what is the effect of EMG signal, ECG signal, BR and GSR signal data collected by various physiological sensors on the main work of this paper, and what is the relationship between them and the PIs in the following paper. These questions should be further explained.
As stated in the article, we conducted the study within the EUROBENCH framework (https://eurobench2020.eu/), which main aim is to provide a unified benchmarking framework for robotic exoskeletons. This framework is comprised of a testing facility, experimental testbeds, and their respective measuring devices, protocols and processing algorithms. As also stated in the article, we adapted our objective to the requisites of the framework, hence selecting the experimental protocols suitable to our objectives, adding our HW and processing algorithms where the testbed did not fulfill our purposes. The EMG, ECG, BR, and GSR were devices required by the PEPATO and EXPERIENCE testbeds, while the processing algorithms to derive the PIs are transparent to the users - the methods were protected by the EUROBENCH consortium. We, therefore, cannot provide further details, and if would, we will incur in disclosing information from other groups, most of it protected. We have provided adequate references to guide the reader to the original articles where the testbeds/PIs were presented by the respective authors.
R.4: In 2.3.2, what is the basis for the allowed error value α to be set at 30%? Does this value have any influence on the subsequent comparison? Further explanation of these issues should be given in the paper.
The AC controller uses the α value as the percentage of movement error allowed by the controller for all the joints. The value of α was defined following a trial-error procedure in which we aimed at allowing tracking error yet having guidance towards the kinematic pattern. The value was in line with what was previously reported for the Exo-H3 exoskeleton in stroke survivors [Bortole et al, 2013 - Reference number 41]. Clarifications on this matter were made on line 192.
The exact value of α sets the guidance provided by the exoskeleton and thus might have an influence on the results of the AC and OC conditions. However, it is out of the scope of the article to investigate the exact influence on the results (i.e. via a correlation analysis between α and the outcome variables), but to compare different control approaches. Then, the α value selected allowed us to set the exoskeleton guidance from rigid (TC) to compliant (AC and OC), to compare these different approaches.
R.5: In 2.5.5, "Non-parametric Friedmann and Post-hoc Wilcoxon tests with Bonferroni correction" and "P-value" are not mentioned in the previous article. The concepts and professional terms mentioned first time in the paper should be explained. What is the function and setting basis of P-value.
The statistical analysis followed in this article is the standard one for testing statistical differences between two related outcome variables, provided the distribution of the data does not follow a normal distribution, or the number of samples is low. Hence the p-value (the Type I error) is set to 0.05% as is the common procedure. Furthermore, when multiple comparisons are involved, the Bonferroni correction is also a standard procedure to adjust the p-value to control for Type I errors.
We have read again section 2.5.5 (Statistical analysis) and compared it with manuscripts with similar themes. We conclude that the paragraph we provided in the text is comparable -in some cases identical- to other articles, but some modifications were introduced (between lines 360 and 368). If the reviewer still feels that this is not suitable, we kindly ask for further details on the intended modification requested, along with an article that could serve us as a guiding example.
R.6: In this paper, the exoskeleton joint data and human physiological data under the three control strategies are compared. However, if the above data under the condition of natural gait (with no exoskeleton) are added to the comparison, the analysis of the results of this paper will be more meaningful.
We thank the reviewer for this comment and partially agree with it. It is true that comparing with non-assisted walking would reveal alterations made in EMG, synergies, fatigue, and most of the outcomes of our work. However, the aim was not to reveal the alterations to healthy walking due to the different exoskeleton controllers, which we already investigated in a previous study (and included in the text)*, but to compare the effects on human-exoskeleton interaction of the three controllers tested, acknowledging that any robotic-assisted walking alters walking*.
Besides, the results provided in the manuscript were obtained in a funded project which ended on June 2022. That project allowed us to pay for access and support to the EUROBENCH facility. Nowadays the facility would cause us further expenses, which cannot be covered by any of our running projects.
Lastly, recruiting again the volunteers to acquire non-assisted walking data would lead to no reliable data, since the biological status of the volunteers may have changed across the 6 months that elapsed since the experiments: weight loss/gain, changes in cardiovascular basal signals (breathing, heart rate, etc.), changes in muscular volume/control due to training/sedentarism, etc.
*We provided this information in the introduction (Lines 85 to 87): “It has been shown that the use of ambulatory exoskeletons does not alter muscle coordination, independently of the level of assistance [31]” and in discussion (Lines 496 to 500): “These results are consistent with other studies in which able volunteers walked at lower speeds using an exoskeleton when compared to self-paced slow walking speed (with no exoskeleton) [31]”.
Round 2
Reviewer 2 Report
The authors have carefully modified the content of the paper and the analysis of the simulation results as suggested previously. The richness of the references is improved, the research route is clearer, and the research results are more credible.